# Sexual Dimorphism in Adipose-Hypothalamic Crosstalk and the Contribution of Aryl Hydrocarbon Receptor to Regulate Energy Homeostasis

**DOI:** 10.3390/ijms23147679

**Published:** 2022-07-12

**Authors:** Nazmul Haque, Shelley A. Tischkau

**Affiliations:** 1Department of Pharmacology, Southern Illinois University School of Medicine, Springfield, IL 62702, USA; nhaque34@siumed.edu; 2Department of Medical Microbiology, Immunology and Cell Biology, Southern Illinois University School of Medicine, Springfield, IL 62702, USA

**Keywords:** obesity, sex differences, aryl hydrocarbon receptor, energy homeostasis, xenobiotics

## Abstract

There are fundamental sex differences in the regulation of energy homeostasis. Better understanding of the underlying mechanisms of energy balance that account for this asymmetry will assist in developing sex-specific therapies for sexually dimorphic diseases such as obesity. Multiple organs, including the hypothalamus and adipose tissue, play vital roles in the regulation of energy homeostasis, which are regulated differently in males and females. Various neuronal populations, particularly within the hypothalamus, such as arcuate nucleus (ARC), can sense nutrient content of the body by the help of peripheral hormones such leptin, derived from adipocytes, to regulate energy homeostasis. This review summarizes how adipose tissue crosstalk with homeostatic network control systems in the brain, which includes energy regulatory regions and the hypothalamic–pituitary axis, contribute to energy regulation in a sex-specific manner. Moreover, development of obesity is contingent upon diet and environmental factors. Substances from diet and environmental contaminants can exert insidious effects on energy metabolism, acting peripherally through the aryl hydrocarbon receptor (AhR). Developmental AhR activation can impart permanent alterations of neuronal development that can manifest a number of sex-specific physiological changes, which sometimes become evident only in adulthood. AhR is currently being investigated as a potential target for treating obesity. The consensus is that impaired function of the receptor protects from obesity in mice. AhR also modulates sex steroid receptors, and hence, one of the objectives of this review is to explain why investigating sex differences while examining this receptor is crucial. Overall, this review summarizes sex differences in the regulation of energy homeostasis imparted by the adipose–hypothalamic axis and examines how this axis can be affected by xenobiotics that signal through AhR.

## 1. Introduction

Obesity, defined as excess accumulation of fat, presents a major risk for several chronic diseases including diabetes, cardiovascular diseases, and cancer. Once considered a problem only in developed countries, obesity has now grown to epidemic proportions across the globe, particularly in urban settings, contributing to millions of deaths each year. Moreover, the obesity problem is not restricted to adults; according to the World Health Organization (WHO), the prevalence of this disease has increased more than four-fold (from 4% to 18%) globally among children and adolescents from 1975 to 2016. An imbalance between energy intake and expenditure is considered a primary cause of obesity. Hence, understanding energy balance is the key to understanding obesity. Adipose tissue serves a vital function, not only in energy storage but also its dissipation when needed, which is achieved by crosstalk between fat depots and the central nervous system (CNS) [1]. As a fast-acting endocrine gland, adipose tissue provides information such as substrate availability, tissue mass, energy intake, and utilization to the brain. The brain processes these signals and directs peripheral tissues to make necessary changes to maintain energy balance. The neuroendocrine signals for energy balance sent by the brain can be either catabolic or anabolic in nature and can vary under different circumstances. One factor that contributes to differences in energy regulation is biological sex. Biological females are more susceptible to obesity, while males are more likely to suffer from diseases associated with obesity, such as diabetes or cardiovascular disease. Evolutionarily, sex-dependent differences in survival strategies suggest that males prepare for periods of energy absence by increasing food/energy intake to increase fat stores, whereas females survive by reducing loss of fat stores by decreasing energy expenditure [2]. These initial observations led to the idea that females might preferentially store fat in subcutaneous adipose tissue (SAT), whereas males utilize visceral adipose tissue (VAT), because subcutaneous adipocytes are more adapted to long term-storage, while visceral adipocytes are more metabolically active [3,4]. Energy storage increases only when intake surpasses expenditure. Adaptive thermogenesis, basal metabolism, and physical activity determine an individual’s energy expenditure. The production of heat from body fuels such as triglycerides in response to environmental change occurs in brown adipose tissue (BAT), which contains vast numbers of mitochondria, and plays a pivotal role in thermogenesis and ultimately, energy balance. Thus, adipose tissue can be categorized mainly into two types: lipid storing white adipose tissue (WAT) and lipid burning BAT. Recently, the recognition of a third type of fat, known as beige or brite (brown in white), has attracted much interest in the metabolic field. It has been postulated that WAT under favorable stimuli/conditions can become thermogenic, which is typically a characteristic of BAT. This causes an increase in energy expenditure through the burning of lipids within the WAT depots, resulting in loss of adiposity [5,6]. Sexual dimorphism is not only restricted to energy storage in WAT, but also found in BAT. Females have more active and larger amounts of BAT [7], as indicated by increased mitochondrial numbers and cristae density [7]. Interestingly, during periods of starvation, females have enhanced capacity to deactivate non-shivering thermogenesis to use energy more efficiently [8]. Collectively, this information suggests that male and female adipose tissues are very different, and therefore, may signal the brains in entirely distinctive ways (Figure 1). Therefore, this review discusses important processes and pathways contributing to sex differences for controlling energy balance and adiposity by exploring crosstalk between the CNS and adipose tissues. The second goal of this review is to discuss environmental modulation of the brain–adipose axis to control energy homeostasis. Although 25–70% of obesity risk derives from genetics [9,10], the remaining 30–75% is sporadic [11]. Identifying specific genes that affect energy balance in the body remains elusive [12]. For the last couple of decades, our lab and others have investigated the metabolic role of the aryl hydrocarbon receptor (AhR), which is perhaps best understood for its function in providing a major defense from environmental toxicants, as well as in regulating development signals. Epidemiological studies link AhR activation by persistent organic pollutants (POPs) to insulin resistance [13,14,15,16,17,18]. Although the mechanisms are not well understood, the highly lipophilic nature of POPs suggest a connection with adipose tissue. In addition, various dietary fats and fat derivatives [19], which deposit in adipose tissue, as well as in the brain, can act as ligands for AhR. This review further explores the relatively new area of AhR research, as an energy balance/metabolism modulator.

## 2. Adipose–Hypothalamic Axis

The adipose–hypothalamic axis contributes a significant role in maintaining energy homeostasis, controlling both energy intake and expenditure. Body fat mass can be a classic example of the feedback signals arising from the fat depots and can be sensed by the brain for maintenance of body weight. Leptin from adipocytes and insulin from the pancreas circulate in the blood in proportion to fat mass and regulates the set point for body fat stores by informing the CNS, acting as adiposity signals to inhibit food intake [1,20]. These hormones act through the hypothalamic melanocortin pathway (Figure 1) [21]. The ARC of the hypothalamus possesses two distinct classes of neurons. One of them secretes pro-opiomelanocortin (POMC), which contributes to the formation of anorexigenic peptides, whereas the other secretes orexigenic peptides, neuropeptide Y (NPY), and agouti-related protein (AgRP). These subsets of neurons are reciprocally regulated by leptin and insulin to reduce appetite and increase energy expenditure, by making numerous connections with other hypothalamic nuclei, such as the lateral hypothalamus (LH), the paraventricular nucleus (PVN), the ventromedial nucleus (VMN), and the dorsomedial nucleus (DMN) [21]. POMC can be post-translationally modified to produce several smaller, biologically active products, including the melanocortins, α, β, and γ melanocyte stimulating hormone (MSH) [22,23]. Melanocortins interact with a family of five G-protein coupled, melanocortin receptors (MCRs) (MC1R, MC2R, MC3R, MC4R, and MC5R) [24,25]. MC3R and MC4R, expressed in the ARC and PVN, are central to the control of body weight [21]. Specifically, activation of these receptors mediates the anorexigenic effects of leptin and insulin.

Circulating leptin and insulin concentrations and sensitivity vary dependent on fat mass, and sex differences are apparent. Secretion of insulin and leptin are highly correlated with amounts of VAT and SAT, respectively [26]. Leptin concentrations are four times higher in women than in men [27,28,29]. The cause is not totally clear but may involve sex steroids. Androgen decreases leptin concentrations in men [30,31], whereas estrogen increases leptin in women [32]. In women, adipose distribution and adipocyte size appear to correlate with leptin levels, which is not the case for men [27,29]. Larger adipocytes and more SAT (not VAT) correlate to higher leptin concentrations in women [33,34]. Significantly higher leptin levels may accelerate the potential for leptin resistance in females compared to males. On the other hand, greater amounts of VAT, together with a lack of protective effect of estrogen, may induce higher insulin resistance in male. Moreover, the female rat brain is more sensitive to leptin, while brains of male rats are more sensitive to insulin (Figure 1) [26]. POMC neurons in the ARC possess receptors for both leptin and insulin, which suggests that POMC neurons can integrate disparate signals originating from the two hormones. The melanocortin system is a common downstream target for both leptin and insulin (activating MC3R and MC4R). Reports demonstrate no sex differences when the agonist (MTII) of MC3 and MC4 receptors administered over a wide range of doses [26]. The sexual dimorphism lies with the input to the melanocortin system, insulin in males, and leptin in females. Thus, the sexual dimorphism of the upstream major mediators of MC3 and MC4 receptors will be discussed in this review.

### 2.1. Sexual Dimorphism in POMC Regulation of Energy Balance and Adiposity

Sexual asymmetry is apparent in the organization of the POMC system. Males exhibit decreased POMC neuronal fibers and projections, as well as reduced levels of the POMC gene and protein compared to females (Figure 1) [35]. Thus, POMC neurons are less active in males, which promotes increased energy intake [36]. The neonatal testosterone surge in males shapes POMC neuron innervation patterns for hypothalamic feeding circuits [35]. Neonatal androgenization of female mice reduces POMC expression and decreases POMC neuronal projections, mimicking a male pattern [35]. These mice display patterns of energy intake and reduced adipose tissue accumulation similar to control males [35]. Although sex steroids are important for establishing sexual dimorphism in the POMC system, many autosomal genes can also contribute in ways that are dependent or independent from sex chromosome gene expression [37]. Some genes can also be a transcriptional target of sex hormones, whereas some could regulate levels of sex steroids or receptors [38,39,40]. For example, STAT3 can be a direct regulator of estrogens and can impart anorexigenic effects in female mice but not in male [41]. Alternatively, genes that are not related to sex hormones/receptors regulation, can still contribute to the sexual dimorphism observed in the POMC neurons, such as TAp63, Sirt1, GABAB [37].

POMC regulation of BAT which controls the body’s non-shivering thermogenesis is also sex divergent [42]. Cholinergic preganglionic sympathetic neurons within the intermediolateral nucleus of the thoracic spinal cord are directly innervated by ARC POMC neurons, and the postganglionic neuron innervates BAT [43,44,45]. Inactivation of ARC POMC neurons can be associated with increased lipid accumulation in BAT and reduced expression of thermogenic genes (*Pgc-1a* and *Elolv3*) in both males and females [42]. Bruke et al. support sexual dimorphism in 5-hydoxytryptamine/serotonin receptor (5-HTCR) expressing neurons within ARC POMC, leading to differences in total energy expenditure, thermogenic activity of BAT, and adiposity [42]. Both male and female mice lacking POMC have increased food intake, reduced thermogenesis by BAT, and decreased physical activitywhich leads to overall increase in fat accumulation and adiposity. The sex discrepancy was revealed when POMC function was restored only within 5-HT2CR expressing cells, where males returned to a lean and healthy metabolic state, but the same transformation was not seen in females. Similar results have been displayed by other labs for male mice. Global deficiency of 5-HTCR develops a late-onset hyperphagic obesity, which is exacerbated by HFD feeding [46,47], and is mostly mediated via the POMC neurons [48,49]. 5-HTCR depolarizes POMC neurons by acting on transient receptor potential channel 5 (TRPC5) [50]. Both 5-HTCR and TRPC5 can be influenced by various hormones on POMC neurons including estrogens, leptin, and insulin [50,51,52,53]. Deletion of TRPC5 from POMC neurons also leads to obesity in male mice due to increased energy intake and decreased energy expenditure [50]. Unfortunately, TRPC5 deletion on female animals and repercussions on energy balance have not been reported. Altogether, these data support sexual dimorphism in 5-HT2CR regulation in hypothalamic POMC neurons to regulate energy balance and adiposity. It is noteworthy that many of the pieces/factors in this story are still lacking due to the purposeful lack of female animals in most of the studies to simplify and reduce efforts.

### 2.2. Sexual Dimorphism in AgRP/NPY Regulation of Energy Balance and Adiposity

Orexigenic neurons in the ARC secrete AgRP and NPY, which act as antagonists of both MC3R and MC4R [54,55,56]. AgRP and NPY application elicits robust hyperphagia and weight gain in rodents, thus linking them to control of eating and body weight [54,55,57,58]. A few compelling studies showed sex differences in both AgRP and NPY in the regulation of energy balance. Central administration of AgRP in both males and females induces robust hyperphagia. However, the effects were shorter-lived in males (Figure 1). Although both groups gained similar amounts of weight, females displayed greater reduction of energy expenditure [59]. Moreover, energy expenditure in females was normalized upon removal of the gonads without any effects on food intake, suggesting that sex differences generated by AgRP are due to sex hormone-specific changes in energy expenditure [59]. NPY also displays sex-dependent differences. Male rats express more NPY mRNA-containing neurons in the rostro–caudal ARC compared to females, which stimulates more food intake by inhibiting the melanocortin system (Figure 1) [60]. Testosterone stimulates NPY expression in ARC nuclei in males [61]. In females, estradiol inhibits the excitability of the NPY neurons in the hypothalamus and stimulates anorexigenic action [62,63,64]. Moreover, overexpression of NPY in adrenergic and non-adrenergic neurons in CNS leads to increased fat accumulation in males, but not females [65]. Altogether, AgRP/NPY and their interactions with various other proteins in regulation of energy balance is complex. The few studies that have been performed show important sex differences, suggesting that additional research is necessary to understand feeding control in males and females.

## 3. Hypothalamic–Pituitary–Adipose Axis

Adipose tissue can be regarded as fast-acting endocrine glands under the control of the traditional hypothalamic–pituitary axes [66]. Adipose tissue expression of specific receptors associated with various hypothalamic–pituitary axes, such as androgen receptor (AR), estrogen receptor (ER), adrenocorticotropin receptor (ACTH-R), growth hormone receptor (GH-R), suggests that in addition to their traditional functions, these axes also regulate adipose tissue function [66,67].

### 3.1. Hypothalamic–Pituitary–Gonadal (HPG) Axis

The HPG axis governs processes that regulate the production of sex-dominant hormones. Organizational, sex-specific effects of the HPG axis arise early in mammalian development. Prenatally, in humans and rodents, the differentiated testis secretes testosterone which creates masculinizing effects, particularly in the brain, where sex hormones affect the organization of neural circuits [68,69,70,71]. This early testosterone surge is crucial for establishing sexual dimorphisms that occur in later life. Specific hypothalamic neurons release the gonadotropin-releasing hormone (GnRH) into the hypophyseal portal circulation in a pulsatile fashion to induce gonadotropin secretion by the anterior pituitary gland. Gonadotropins govern the production of steroids, such as estradiol, progesterone, and testosterone by the gonads. Males and females produce sex steroids in different concentrations with sex-specific physiological consequences. In females, the main circulating estrogen is 17β-estradiol (E2), although lower levels of estrone (E1) and estriol (E3) are also present. Estrogens bind typically to either estrogen receptor alpha (ERα) or beta (ERβ), which have organ-specific distributions, and act as transcription factors to regulate gene expression [72]. Additionally, estrogens can also bind a membrane-associated G-protein coupled estrogen receptor (GPER) to initiate non-genomic biological effects [73]. Similar to estrogens, the body produces several androgens that can act differently depending upon their target [74]. Dihydrotestosterone (DHT) and testosterone (T) are the most potent androgens and impart their biological effects binding to the AR, which also acts as a transcription factor to regulate gene expression [75]. Moreover, circulating T can be converted to estrogen by the aromatase (CYP19A1) enzyme. Steroid receptors are found in both WAT and BAT [76], where they can contribute in sexual dimorphism to regulate energy homeostasis. Since androgen is the main circulating sex hormone in men and estrogen in female and both are considered the most critical sex steroids in adiposity, this review will be focused mainly on these two hormones.

#### 3.1.1. Estrogen Regulation of Adipose Tissue for Involvement in Sex Difference Energy Homeostasis

In humans and laboratory animals, estrogen regulates energy balance through effects in WAT in both sexes, but especially in females [77,78]. Low estrogen levels increase WAT, whereas estrogen supplementation decreases WAT [79]. Although ERα and ERβ are expressed in WAT [80], ERα predominates [81]. ERβ may regulate metabolic function, such as insulin sensitivity and glucose tolerance, but is not compelling for fat deposition and energy balance [82,83]. Both male and female VAT have increased ERα receptor compared to SAT. E2 increases the ERα levels in these tissues [84]. Many of the protective effects of estrogens against adiposity are likely mediated through ERα in both sexes [81,85,86]. In females, ERα inhibits WAT development as well as the amount of WAT (Figure 1) [86]. ERα signaling in WAT affects adipocyte size [86,87], which suggests that estrogen regulates triglyceride accumulation. In adult females, estrogen deficiency enlarges adipocyte volume, whereas both number and size of adipocytes increases after estrogen depletion in juveniles [88], indicating that estrogen signaling may impact differentiation and/or proliferation of the adipocyte lineage. In males, the role of estrogen in regulating WAT volume and size remains unclear. Moderate increases of both WAT size and volume occurs in αERKO male mice, but whether these effects are strictly due to estrogen, or to other hormones, is not certain. In addition, estrogen can increase the level of proteins like Heat Shock Protein (HSP) 72 [80,89], adipokines like adiponectin [81], and glucose transporter 4 (GLUT4) [90,91], which can contribute to protection from insulin resistance. E2 may also regulate lipogenesis and lipolysis functions of adipocytes. Though some controversy remains, the majority of studies suggest that estrogen reduces lipogenesis [92,93], and increases the lipolysis rate [94,95], at least in females. Physiologically, ERα-mediated suppression of WAT deposition may reflect increased energy expenditure in females [86,96,97,98].

Estrogens are also critical regulators of BAT activity, especially in females [99]. Because estrogen affects energy expenditure rather than in energy intake, the thermogenic function of BAT is important to consider. Proliferation, differentiation, and thermogenic activity of BAT are all regulated by ERα signaling [100,101,102]. ERα transcripts are higher in female BAT, suggesting an important ERα role in regulating energy expenditure [84]. Thermogenic activity of BAT in female mice is reduced after ovariectomy [103,104], and can be restored after systemic administration of estradiol (Figure 1) [103,104,105]. In males, fat burning capability is correlated with the aromatization of testosterone (T) to estrogens and concurrent ERα signaling, which might occur in all male fat depots, and not necessarily BAT [106]. Although the precise mechanism remains elusive, inhibition of aromatase in testosterone-treated hypogonadal males increases fat mass, suggesting that the effects of T are mediated through its local conversion to estrogen [107]. Overall, the studies indicate estrogen signaling in adipose tissues, mediated by ERα, is controlled by the HPG axis which imparts a protective effect against excessive positive energy and adiposity, mainly in females with some extent to males depending upon aromatization of T.

#### 3.1.2. Androgen Regulation of Adipose Tissue Sex Differences in the Involvement of Regulating Energy Homeostasis

Similar to ER, AR is also expressed widely throughout adipose tissue (WAT and BAT) in both sexes [108]. AR transcript levels are significantly higher in VAT in contrast to SAT [84]. These differences in conjunction with T levels in males further explains the higher responsiveness of VAT to androgens compared to SAT. Masculinization or defemination by androgens can significantly impact metabolism, and can reprogram the genetic predisposition towards food intake, obesity, and metabolic dysfunction [35,109,110,111,112,113], creating sex differences in adipose tissue physiology and regulation of energy balance [35,109,114]. Androgen deficiency decreases food intake and reduces weight gain [115]. In males, androgens can prevent obesity by regulating the deposition of fat [116]. In hypogonadism, low T is associated with an increased amount of VAT [27,117], which can be decreased by T replacement in a dose-dependent manner (Figure 1) [118]. T increases lipolysis in male and female SAT, but not in VAT [119,120]. T concentrations that are 15-fold higher in adult males compared to females support the observation that males accumulate VAT [121]. Lipogenesis is also sexually dimorphic in VAT, where the process is reduced in males and increased in females [122]. However, androgen effects on lipogenesis are not clear. One study in castrated male nonhuman primates found no effects of T on lipogenesis [123]. Furthermore, androgen effects in BAT remain unclear and understudied. T treatment of cultured brown adipocytes reduces expression of mitochondrial biogenesis genes and increases lipid accumulation [101,124]. In contrast, orchiectomy or AR global knockout also reduces mitochondrial biogenesis (high UCP1 mRNA levels) and thermogenesis [125,126] in BAT [127]. DHT does not affect Ucp1 mRNA expression in orchiectomized mice [128,129,130]. It is likely that differences in androgens, and perhaps receptor expression, may underlie these divergent results. Moreover, interactions of other hormones, including estrogens and catecholamines, together with T, must be explored.

Androgens can also affect female BAT. In the hyperandrogenic state associated with polycystic ovarian syndrome (chronic androgen excess), females experience abdominal obesity and metabolic disorders [131]. Reduced BAT activity, as demonstrated by decreased thermogenic gene expression in BAT may contribute. Overall, the studies implicate androgens in regulating energy metabolism in both sexes through regulating BAT activity. Both ERα and AR are high in BAT and changes in steroid receptor expression and thus the sex hormones can affect the activity of BAT, which requires further investigation.

### 3.2. Hypothalamic–Pituitary–Adrenal (HPA) Axis

Stress invokes the body’s defenses in response to a threatening situation, including altering metabolism. Acute stress, which is mediated by the HPA axis, is commonly associated with reduced food intake and weight loss [132]. Stress leads to the release of the corticotropin releasing hormone (CRH) from the PVN of the hypothalamus, which subsequently stimulates the release of adrenocorticotropin (ACTH) from the anterior pituitary. ACTH in turn triggers the production and release of glucocorticoids from the zona fasciculata of the adrenal cortex. Glucocorticoids act through two types of receptors, the glucocorticoid receptor (GR) and mineralocorticoid receptor (MR), to exert its action in several tissues. Glucocorticoid receptors in the HPA mediate negative feedback to suppress the stress response after glucocorticoid levels rise [133]. CRH induces anorexia in the brain by inhibiting neuropeptide Y (NPY)-stimulated food intake [134]. Moreover, CRH can also stimulate the sympathetic nervous system (SNS) and production of catecholamines to further reduce food intake, promoting lipolysis and BAT thermogenesis [135,136,137]. Catecholamines, such as epinephrine and norepinephrine (NE), act through α(1–2) and β(1–3) receptors [138] expressed in central and peripheral targets, including adipose tissue, to coordinate the ‘fight or flight’ response and impact metabolism. For example, thermogenesis in BAT is mainly mediated by NE activation of β3 adrenergic receptors through the SNS [139]. Collectively, the SNS and HPA axis allows adjustment of metabolic needs to survive during temporary threatening situations. Complications arise when CRH is produced for longer periods during chronic stress, as that leads to increased production of glucocorticoids and dysregulation of negative feedback [140,141,142,143]. Chronic stress leads to development of visceral obesity and other metabolic complications [144,145]. Chronic stress can also potentiate food intake through a series of complex interactions, including upregulation of orexigenic NPY/AGRP expression in the hypothalamus ARC nucleus [146]. Furthermore, chronically stressed animals prefer calorically dense food, as it helps ameliorate HPA axis response to further stress by activating the food reward system [147,148,149]. In addition, chronic glucocorticoid elevation promotes the release of anorexigenic hormones, leptin and insulin, which ultimately leads to leptin and insulin resistance, thus reducing their ability to induce satiety effects in the brain [150,151,152,153]. Glucocorticoids also play an important role in the regulation of lipid homeostasis in adipose tissue. Abundant glucocorticoids receptors in VAT selectively promote fat accumulation compared to other adipose depots [154,155,156,157].

Sexual dimorphism in metabolism incurred by chronic stress remains under-reported. Nevertheless, the consensus is women are more likely to develop obesity under chronic stress conditions [158,159]. As opposed to males, glucocorticoids have stronger effects on SAT in females [160]. In females, glucocorticoid stimulation of lipolysis is enhanced in SAT compared to VAT, whereas no differences were found between the depots in males [160]. This may partially explain why females tend to accumulate SAT and are less susceptible to obesity-related metabolic complications. 11β-HSD1, an enzyme that increases the conversion of active glucocorticoids from inactive forms, is more prevalent in males, and is overexpressed in VAT, which may contribute to the accumulation of VAT in chronically stressed males compared to females [161,162,163,164]. Sexual dimorphism in activity of catecholamines to modulate lipolysis and adiposity may also play a role. Probably due to larger adipocyte size and more lipoprotein lipase activity, males have higher lipolytic activity in VAT, which contributes to more lipid mobilization. Moreover, males also have increased sympathetic neuronal projections and postsynaptic adrenergic receptors in VAT [165,166]. Glucocorticoids can also suppress BAT activity, by downregulating thermogenic gene expression [167]. Corticosterone treatment not only increases lipid accumulation, but also decreases catecholamine-induced UCP1 expression in BAT in both male and female mice [168,169,170,171]. Moreover, elevated 11β-HSD1 in BAT increases the availability of GC within the tissue [172,173]. The inhibitory action of GC is likely mediated through GR, since the antagonist of this receptor blocks GC-induced UCP1 expression [174,175,176]. In contrast, BAT-specific GR knockout male mice do not display changes BAT thermogenesis or HFD-induced metabolic function [177]. GC levels and the HPA axis function is unaffected in BAT-specific GR knockout male mice. However, adrenalectomy, which reduces both GC and catecholamine levels, induce BAT activity in male mice [170]. Collectively, although GC effects on BAT remain under investigated, nevertheless HPA axis regulation in this depot surely occurs. Studies in female mice are limited, yet one can hypothesize that stress might also play an integral role in BAT to influence obesity by observing the greater preponderance and sensitivity of the environment towards this depot in females. More studies that explore sex-specific effects are warranted. Thus far, it appears that females have higher responsiveness to stress stimuli compared to males regarding regulation of energy homeostasis (Figure 1). Moreover, specific adipose tissue depots play specific roles to affect the consequences of HPA activity.

### 3.3. Hypothalamic–Pituitary–Somatotropic (HPS) Axis

Under regulation of the hypothalamic growth hormone-releasing hormone (GHRH) and somatostatin peptides, the growth hormone (GH) is secreted in a pulsatile fashion by the anterior pituitary gland. GH release is stimulated by GHRH, and reciprocally inhibited by somatostatin [178,179]. GH secretion is influenced by age, gender, food, circadian rhythm, body fat composition, and other factors [178,179]. Numerous physiological processes including somatic and bone growth, energy balance, body composition, glucose and lipid metabolism are regulated by GH. Adipose tissue is a major target of GH, expressing an abundance of GH receptors (GHR) to regulate key functions such as proliferation, differentiation, and lipolysis [180,181,182]. GH overall imparts positive effects in adipose tissue, and mice lacking GHR have impaired development of adipose tissue, with dysregulated differentiation and proliferation [183,184]. It directly stimulates lipolysis, inhibits lipogenesis, and reduces fatty acid synthesis in adipose tissue [182,183,185]. GH also contributes to nutrient partitioning and energy balance by favoring muscle growth, protein synthesis, and reducing fat mass [183]. Obese individuals have reduced GHR expression in adipocytes and the spontaneous pulsatile pattern of GH secretion from the hypothalamus is markedly decreased, suggesting that GH may have significant impact in the development of obesity [186,187,188,189]. GH secretion is sex specific, with lean mean 24-h GH release and levels elevated in females [190]. Both testosterone and estrogen increase GH secretion through different mechanisms [191], although estradiol has greater influence on pulsatile GH release [192]. Sex steroids also regulate GH receptor expression in various tissues, including adipose [193]. VAT is the most crucial factor in regulating the 24-h GH release in lean adults, suggesting again a pivotal role for the brain–adipose axis in homeostatic physiology. For each increment of VAT, there is an exponential decrease in the 24-h GH level [194]. Weight loss can fully reverse hyposomatotropism associated with morbid obesity [189,195], however, GH supplementation does not treat obesity [187]. This suggests reduced HPS activity is a consequence rather than cause of obesity. Although underlying mechanisms remain to be elucidated, it seems that VAT sends a negative feedback message to the GH axis. Elevated levels of FFA, insulin, and IGF-1 due to adiposity may also contribute to the inhibition of GH secretion [196,197,198]. There is an inconsistency among various studies conducted between sexes with respect to body fat and GH release, yet the data are more robust in men [190,199,200]. Men typically have a lower overall percentage of body fat, albeit a higher amount of VAT, contributing to sex-related variation in adiposity. Understanding the relationship between fat signaling and GH release in the context of different sexes requires further investigation.

## 4. Xenobiotics Modulating Aryl Hydrocarbon Receptor to Regulate Energy Homeostasis

The fact that industrial chemicals adversely affect human health is not new; environmental contaminants can have a profound, long-lasting impact on numerous disease conditions, including obesity and metabolic dysfunction. The aryl hydrocarbon receptor (AhR) is a principal target for neutralizing a variety of environmental toxicants. Moreover, AhR contributes to several physiological functions in the brain, which includes neuroendocrine functions, neurogenesis, cell differentiation, and cell survival [201]. AhR has been characterized in several animal models [202]. This receptor is present in the cortex, hippocampus, cerebellum, and highly expressed in the hypothalamus when compared to other regions of the brain [203]. AhR is a basic-helix-loop-helix/period-aryl hydrocarbon nuclear translocator-single minded (bHLH/PAS) family of genes, which binds a variety of ligands. bHLH-PAS proteins typically possess two separate PAS domains, PAS-A and PAS-B, composed of 50 amino acid repeats that mediate protein–protein interactions. Homomeric or heterodimeric PAS proteins act as transcription factors, which bind to DNA through the bHLH domain to affect expression of many target genes and numerous physiological functions [204,205,206]. Functions of AhR are influenced by the ligand affinity, heterodimeric partnerships, cell-type, and other environmental factors. Canonical AhR signaling starts from ligand binding which causes the receptor to be translocated inside the nucleus, where it complexes with the AhR nuclear translocator (ARNT) and forms a heterodimer. The heterodimer then binds to the xenobiotic response element (XRE) sequence to regulate transcription of various genes including the cytochrome P450 Cyp1 family, phase II detoxification genes, and numerous others [207], including genes contributing to the endocrine system [207,208]. AhR also interacts with various members of the nuclear receptor superfamily, including receptors for estrogens [209,210,211], androgens [212,213,214,215], glucocorticoids [216], and thyroid hormones [217]. AhR-ER crosstalk was first proposed around four decades ago and is complex. A majority of studies support inhibitory interactions, where activated AhR attenuates the ERα signaling. The precise mechanisms remain equivocal, but some evidence supports metabolism of estrogen via the induction of cytochrome P450 Cyp1 family to increase proteasome-mediated degradation of ERα, and recruitment of ERα by AhR ligands to the AhR bound promoters, reducing the ERα signaling [218]. AhR can also be anti-androgenic, through interference between AhR ligands and transcriptional interference with AR in testosterone signaling pathways [212]. Moreover, AhR is expressed in all cell lineages of pituitary tissues in both mice and humans [219,220,221]. Most works on AhR actions in pituitary have focused on endocrine disruption and xenobiotic effects. It is well accepted that AhR signaling is crucial for sex-steroid biosynthesis during the fetal period, which helps determine sexual dimorphic phenotypes detailed below in the next sub section. In the context of energy homeostasis, pituitary hormones have XRE sequences in their promoters, which indicate that AhR may affect their expression. For example, POMC, which is a precursor protein not only for α-MSH (that regulates appetite) but also for ACTH, has several upstream XRE motifs [222]. GH also possesses XRE sequences in its promoter, for which AhR can compete to regulate its expression [223,224]. Beta-naphthoflavone (BNF), an agonist of AhR, can disrupt several genes involved in the neuroendocrine regulation of stress [225,226]. Overall, most studies have shown AhR to affect the HPG axis, while the effects on GH, TSH, and ACTH are still not clear [227,228,229,230]. This section highlights key discoveries related to AhR influences on the neuroendocrine system to control energy homeostasis and provides rationale for exploiting AhR as a therapeutic target in obesity for both the sexes.

### 4.1. AhR in Early Sexual and Neuroendocrine Development

Emerging evidence suggests that early life exposure to environmental toxicants can have long-lasting impacts on development and health. bHLH-PAS family proteins are crucial regulators of the hypothalamus and neuroendocrine development [231,232,233]. As a mediator of toxicity to environmental toxicants, AhR can be activated by numerous exogenous ligands originating in air, earth, water, and living organisms. Most toxic contaminants that are AhR ligands are man-made and are produced by various industries, including pesticide, bleaching, wood preservation, metallurgy, and many more. Furthermore, naturally occurring and endogenous ligands can activate AhR and influence physiological function. Among the man-made ligands that generate major health concerns are halogenated aromatic hydrocarbons, such as polyhalogenated dibenzodioxins, dibenzofurans, biphenyls that bind to AhR with high affinity even in the pico-nanomolar range [234]. Many exogenous AhR ligands such as TCDD and PAHs can cross the blood brain barrier (BBB) to mediate AhR action in the brain. Likewise, many endogenous AhR ligands derived from tryptophan metabolites such as indoxyl-3-sulfate (I3S), indole-3-carbinol (I3C) and FICZ also cross the BBB via gut–brain axis to modulate AhR activity in the brain in response to various environmental and metabolic cues [226,235]. Their ubiquitous distribution, lipid solubility and long-lasting half-life, promote bioaccumulation of the compounds throughout the food chain by depositing in lipid-heavy tissues such as adipose and brain [236,237,238]. There they show immense resistance to breakdown. The Environmental Protection Agency (EPA) has recognized a link between an increased morbidity rate and level of Polychlorinated Biphenyls (PCBs), which also includes dioxins, in the general US population [239]. Alarmingly, people in highly contaminated regions might have surpassed the tolerable dioxin exposure [240]. Moreover, dioxins may accumulate during prenatal and postnatal periods via the placenta and breast milk [241,242,243]. With a half-life that is roughly decades, the chances of staying in tissue from birth to adult to activate AhR chronically are high. Dioxin burdens in the perinatal stage can lead to irreversible changes in brain development that become apparent in adulthood. Therefore, understanding how these toxicants impact development is critical. Reproductive development appears sensitive to toxicant exposure. Prenatal exposure to the potent AhR agonist, TCDD feminizes male rats [239,244,245,246]. TCDD-treated male mice exhibit gonadotropin secretion patterns similar to females, and decreased plasma androgen levels [245]. Feminization of the preoptic area of the hypothalamus may contribute to this feminization [247]. Prenatal exposure to TCDD in females leads to reproductive dysfunction in adulthood, including complications in estrus cycle, ability to achieve and maintain pregnancy, and sometimes infertility (Figure 2) [244,248,249]. Altered gonadotropin release patterns due to TCDD-induced alterations of the POA is considered to be the reason for diminished reproductive capacity.

The link between AhR and sex steroid receptor pathways prompted exploration of AhR expression in known sexually dimorphic areas of the brain. Interestingly, AhR and ARNT expression is sexually dimorphic in the hypothalamus, particularly in the POA, which is important for sex behaviors [250,251]. Overlap with regions high in ER [252] may explain why the alteration of gonadal hormones due to TCDD exposure leads to sexual dimorphism. Similarly, interaction between AhR and ER may affect function of other hypothalamic nuclei, including the anteroventral periventricular (AVPV), arcuate (ARC), and ventromedial (VMH). Interestingly, these regions regulate both sexual behavior and energy homeostasis [253,254,255,256]. Manipulation of sex hormones, and their receptors during development can lead to permanent changes in neuronal connectivity and functions, and exogenous AhR ligands may affect perinatal neuronal development, which can be irreversible. Therefore, it is very important to understand the molecular mechanisms related to AhR action in the developing brain to evaluate the possible impact on human health.

### 4.2. AhR Contribution to Energy Metabolism and Obesity

Toxicological studies on AhR led to the hypothesis that AhR might regulate energy balance [257]. Among many symptoms, TCDD toxicity in humans causes anorexia and weight loss [257]. Targeted studies revealed that AhR can modulate metabolically important gene expression, including regulation of blood glucose, lipid, and energy homeostasis [258,259,260,261]. Initial studies in mice deficient in AhR directed a focus on metabolism due to developmental defects involving liver and other metabolically important organs. However, the first evidence of AhR regulation of energy metabolism appeared a decade ago [262,263]. Lee et al. demonstrated that activation of AhR has the potential to cause fatty liver disease/hepatic steatosis through altered fatty acid transport inside liver, and increased lipid spillover from fat depots. Our lab demonstrated a direct connection between AhR deficiency and peroxisome proliferator-activated receptor alpha (PPARα) loss of function in liver to hinder glucose and fatty acid metabolism. It is now well established that AhR deficiency protects mice from high-fat diet (HFD)-induced obesity [264,265]. Under HFD, tryptophan metabolites that act as AhR ligands are increased, which activate AhR and promote an obese phenotype [266]. WAT is critical for metabolism of kynurenine (Kyn), a downstream catabolite of Trp; HFD increases circulating Kyn in obese individuals [267]. Exhaustion of Trp inhibits the production of other Trp metabolites such as serotonin, which is involved in satiety. Moreover, excessive Kyn promotes AhR activity to activate the AhR/Stat3/IL-6 pathway in adipocytes and mediates the development of obesity and insulin resistance [268]. Our lab found that HFD alters many key hepatic and adipose genes, including fat synthesis, accumulation, and catabolism pathways, which contribute to dietary obesity and insulin resistance. Moreover, AhR deficiency protects against HFD-induced elevations in leptin and insulin, and reductions in adiponectin, which are all indicators of metabolic dysfunction. An increase in energy expenditure in AhR-deficient mice, associated with enhanced expression of uncoupling protein 1 (UCP1) contributes to metabolic protection in these mice. Furthermore, mice with a low affinity form of AhR (B6.D2) are also protected from diet-induced obesity [264]. A variety of nuclear receptors (such as genes from the PPAR family) were important for fat biogenesis, accumulation and mobilization are altered in these animals. Together, this indicates that reduced AhR function provides mice with mechanisms that maintain healthy energy balance in the face of HFD.

AhR can also regulate fibroblast growth factor 21 (FGF21), which protects properties against metabolic disease by promoting energy expenditure and improving both lipid and glucose metabolism [269,270]. Produced by the liver, circulating FGF21 acts on adipocytes to promote thermogenesis, insulin sensitivity and produce a favorable lipid profile [271,272,273]. FGF21 can also be released from adipose tissue to act in an autocrine or paracrine fashion to increase thermogenesis [272,273,274,275]. FGF21 promotes BAT thermogenesis and differentiation of WAT to produce increased numbers of *brite* adipocytes (also known as browning of adipose tissue), through induction of PPARγ [276,277]. *Brite* or *beige* adipocytes express UCP1 and have brown adipocyte-like function [278]. Dense caloric intake stimulates FGF21 expression through a PPARγ in an attempt to improve insulin sensitivity and adipocyte function [279]. Noradrenaline is a potent activator of UCP1 expression and induces ‘*browning*’ of WAT [280]. FGF21 expression is also increased in WAT and BAT after physical activity or cold exposure, and its release has been correlated with release of noradrenaline from activation of SNS [277,281,282,283,284]. There is clear evidence that AhR regulates FGF21 expression, through XRE regions in the FGF21 promoter. Whether AhR up-regulates or down-regulates FGF21 gene expression remains equivocal, with data to support each effect. Some labs found AhR activation suppresses FGF21, and liver-specific deletion induces its expression in mice [285]. However, other studies indicate TCDD activation of AhR promotes hepatic FGF21 expression [286]. Differences in the duration of AhR activation may reconcile the disparate results. Acute induction of FGF21 by short-term AhR activation might be beneficial, whereas long standing AhR activity may have opposing effects, including development of FGF21 resistance. The developmental stage of AhR deletion may also complicate the issue. Postnatal deletion of hepatic AhR may affect weight loss through increase in adipose-regulated increases in energy expenditure [287]. Protection of mice bearing a constitutively active form of AhR from diet-induced obesity and diabetes is abolished upon FGF21 knockdown [288]. In summary, although the exact consequences of AhR-FGF21 interactions remain unclear, AhR seems to use both the FGF21 and PPAR family to modulate lipid and energy metabolism.

### 4.3. Sex-Specific AhR Modulation of Energy Balance

Obesity and its associated diseases, such as diabetes, can develop through sex-specific mechanisms manifested by dissimilar gene expressions in metabolic tissues [289]. AhR affects energy homeostasis and gene expression patterns in many metabolic tissues including adipose tissue, liver, and brain, which also display differences between sexes. AhR contributes to sex-specific differences in a complex manner. Effects of AhR deletion appear obfuscated by the timing of AhR deletion. For example, in liver, conditional knockout during gestation shows different effects [290]. Specific gestational deletion of AhR from liver exacerbates metabolic disease conditions, such as hepatic steatosis, under HFD, whereas CKO from adult liver helps ameliorate the pathology [287,290]. Molecular studies substantiate these findings, where HFD treatment of animals with gestational excision showed augmented or unchanged gene expression related to various metabolic processes; there was an increase in lipogenesis and inflammation, and no differences in fatty acid uptake, β-oxidation, or gluconeogenesis. On the other hand, adult CKO of AhR from liver demonstrated significantly less weight gain and adiposity from HFD. These animals had increased respiratory capacity of BAT and WAT, due to more production of FGF21 by the liver. Our lab obtained similar results in an adipose-specific adult CKO (Haque and Tischkau, unpublished results) mouse compared to gestational adipose-specific AhR CKO [291]. Gestational AhR deletion from WAT exhibited increased weight gain, adiposity, inflammation, and significant impairment in glucose homeostasis when fed HFD [291]. In contrast, adult AhR CKO from all types of adipose tissue protected males and females gained significantly less weight and were protected from hepatic steatosis when fed HFD. Effects of adult CKO were more profound in females. Female AhR CKO mice gain metabolic protection from FGF21, PPARγ, and ERα pathways. Increased *beiging* of WAT, adipogenesis, and decreased VAT are more dominant in females. It appears that AhR-specific adipose deletion protects females from leptin resistance and males from insulin resistance (Figure 2). Although leptin and insulin are satiety hormones, leptin modulates satiety through numerous pathways, and has more robust effects compared to insulin. Leptin can also be modulated by sex steroids; estrogens enhance leptin sensitivity, whereas testosterone induces leptin resistance [292,293]. Moreover, female brains are more sensitive to leptin, whereas males are more reliant on insulin. The effects on leptin and insulin necessitate a better look at sex-specific control of feeding at the central level. Unfortunately, there are currently no available data on hypothalamic effects of AhR loss in energy balance. Both sexes express significant levels of AhR in the hypothalamic arcuate nucleus [201,247]. These studies are necessary to understand the role of AHR in regulating whole body energy metabolism.

## 5. Conclusions

Until recently, research on many physiological pathways, including energy homeostasis, has focused on males, as a representative of mammalian species. The existence of sex differences has been increasingly appreciated due to many exciting discoveries. As discussed in this review, obesity is fundamentally different between the sexes, with many pieces that remain poorly understood. Genetics, environment, and sex steroids at the prenatal, fetal, and puberty periods define the early and late changes in brain–adipose physiology for both sexes. Neurons and adipose tissue cross talk significantly to influence sex-specific gene networks and cellular systems that determine body composition and mechanisms for energy homeostasis (Figure 1). Additionally, xenobiotics from industrial waste or food is a key environmental factor that can permanently impair neuroendocrine function in males and females. As a xenobiotic sensor that responds to lipophilic ligands, AhR is an interesting target for regulation of energy metabolism in brain and adipose tissue. Both over-activation and inactivation of AhR can lead to dysregulation of physiological homeostasis, and the activity of the receptor can closely be controlled to ameliorate disease pathologies associated with this gene.

Reduced AhR activity systemically can alleviate obesity, although this may result in many unwanted side effects and thus tissue-specific inhibition is more desirable. However, the timeline of such inhibition is extremely important. Gestational inhibition can exacerbate metabolic dysfunction, whereas adult inhibition is more advantageous for the disease condition. Ablation of AhR activity from mature liver and adipose-specific deletion of AhR is helpful in combating obesity. Mechanisms used by both the specific deletions are different and distinguish the specific role played by each of the tissues. Such studies need to be conducted in tissues such as the brain (specifically hypothalamus and pituitary) to delineate the tissue-specific role of AhR in obesity. With this information, we can identify appropriate tissues for developing and designing drugs to target AhR in combating obesity with minimum side effects. Moreover, investigating sex differences while conducting these studies are paramount since as discussed, the role of sex is a fundamental factor in the incidence of obesity.

## Figures and Tables

**Figure 1 ijms-23-07679-f001:**
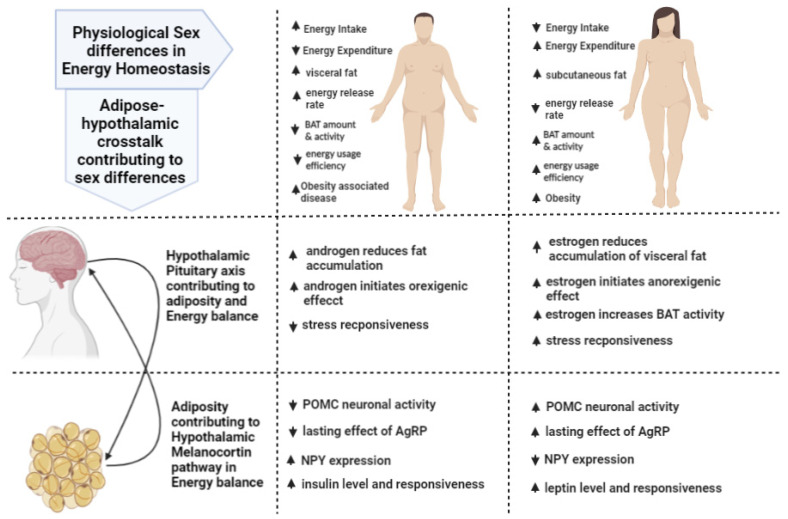
Schematic overview of physiological/pathological sex differences in energy homeostasis regulation and how the adipose–hypothalamic axis contributes in the regulation. Upward or downward arrows indicate higher or lower levels, respectively, of the given phenomena compared to each sex’s counterpart. POMC, pro-opiomelanocortin; NPY, neuropeptide Y; AgRP, agouti-related protein; BAT, brown adipose tissue.

**Figure 2 ijms-23-07679-f002:**
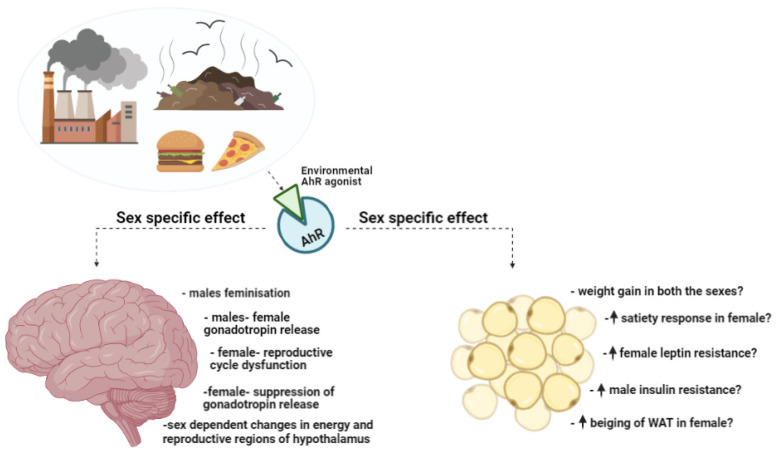
A simplified model depicting crosstalk between the environment and mammalian system through AhR, and how activating this receptor can contribute to sex-dependent changes in the brain (hypothalamus) and adipose tissue to effect energy homeostasis. AhR, Aryl Hydrocarbon Receptor; WAT, White Adipose Tissue.

## Data Availability

Not applicable.

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
