# Peer review of "Sexual Dimorphism in Adipose-Hypothalamic Crosstalk and the Contribution of Aryl Hydrocarbon Receptor to Regulate Energy Homeostasis"

_ijms, 2022, doi:10.3390/ijms23147679_

Round 1
Reviewer 1 Report
The review provided by Haque and Tischkau describes how obesity is a sexually dimorphic disease, emphasizes the importance of a sex-dependent cross talk between adipose and the brain that contributes to energy homeostasis and body composition, and how targeting AhR for treating obesity should be considered in a sex-specific manner since it can modulate sex steroid receptors. The review was thorough, provided simplified diagrams to outline complex mechanisms, and was an enjoyable read.
Major Point:
Based on the title, I was anticipating a thorough discussion related to sexual dimorphism of xenobiotics and their contribution to metabolic disease through AhR; unfortunately, this was only a minor section (4.3) described at the end of the review. Further exploration into this topic would benefit the review, especially discussing pertaining to theories/mechanisms involved in the processes outlined.
Introducing beiging process of adipose tissue earlier into the review, specifically in the introduction would also be a tremendous benefit.
Discussion pertaining to sex-dependent responsiveness to insulin and leptin was included (lines 109-122); however, is there any sex-dependent mechanisms the reader should be aware of related to leptin/insulin resistance development?
Paragraph lines 140-152 outlines sex-specific differences regulating BAT between males and females, specifically the role of POMC 5-HTCR function. Can the authors comment as to how these differences may relate to overall differences in energy homeostasis in mice?
Minor Points:
Remove "the" line 43
Figure 1: "subqutanous fat" mispelled
Placing spaces between the [reference] and words
Run-on sentence lines 129-131; consider rephrasing
Insert "and" between "growth hormone receptor, AND estrogen receptor" line 179
Can the authors comment whether hormone receptor expression differs in adipose tissue of males and females in section 3.1?
Can the authors comment as to any knowledge about the effect of reproductive cycling or menopause in females and whether this has an influence on adipocyte biology or the brain-adipose axis? Do we know whether males and females, when responding to the same stressor, have differences in lipolysis rates?
Figure 2: "beiging" mispelled
Reviewer 2 Report
The manuscript “Sexual Dimorphism in brain-adipose crosstalk and the contribution of xenobiotics modulating Aryl Hydrocarbon Receptor to regulate energy homeostasis” written by Nazmul Haque and Shelley A. Tischkau, aimed to review potential role of AhR in sex differences of metabolism, particularly brain energy expenditure in obesity. Indeed, there was a recent increase in interest in the physiological functions of the AHR. To date, 3915 AhR-binding sites have been identified in the human genome, thus, it is important to elucidate the functions of these genes and their involvement in diseases. Therefore, understanding the role of AhR in energy balance/metabolism is an emerging and interesting problem. Overall, the review is well written and touches upon important biology, however I did have some questions that need to be addressed before publishing this article.
In Introduction: Lines 77-78: ( Epidemiological studies link AhR activation by persistent organic pollutants (POPs) to insulin resistance) - the References are missing. I also have questions regarding Chapter 4: Xenobiotics modulating Aryl Hydrocarbon Receptor to regulate energy Homeostasis. According to recent data, the highest level of AhR expression is recorded in the human placenta, followed by the lungs, heart, pancreas and liver, while it is expressed at a low level in the kidney, brain and skeletal muscle (Zhu K. et al., Aryl hydrocarbon receptor pathway: Role, regulation and intervention in herosclerosis therapy. Mol. Med. Rep. 20, 4763, doi: 10.3892/MMR.2019.10748). Therefore the question arises: how important is the role of AhR in the brain? In my opinion, the review lacks current knowledge about AHR activity in different parts of the brain and omits the discussion on which xenobiotics can cross the blood-brain barrier. The review mentions the data on the effects of TCCD on the mouse brain, but it is not immediately clear how other xenobiotics may work in this context? Most of the data mentioned in this review that addressed the regulation of AHR in mice has been obtained in the last 10–20 years. I would like to see more up-to-date results obtained from human cells and that would include transcriptomic analysis or if such data do not exists – explain what are the probable experimental hurdles that not allow to obtain such results. In my opinion the review also lacks a summary scheme: ligand-AhR-liver, brain, adipose tissue, females, males. It would help to understand the general idea of ​​the review.
